# Efficiently Training Time-to-First-Spike Spiking Neural Networks from Scratch

**Kaiwei Che** [1 2]   **Zhengyu Ma** [2 *]   **Yifan Huang** [3]   **Peng Xue** [2 4]   **Li Yuan** [1 2]   **Wei Fang** [1]   **Yonghong Tian** [1 2 3]

## Abstract

Spiking Neural Networks (SNNs), with their event-driven and biologically inspired mechanisms, are well-suited for energy-efficient neuromorphic hardware. Neural coding, which is critical to SNNs, determines how information is represented via spikes. While Time-to-First-Spike (TTFS) coding uses a single spike per neuron to offer extreme sparsity and energy efficiency, it often suffers from unstable training and low accuracy due to its sparse firing. To address these challenges, we propose a training framework that incorporates parameter initialization, training normalization, a temporal output decoder, and a re-evaluation of the pooling layer. The proposed parameter initialization and training normalization mitigate signal diminishing and gradient vanishing, which helps stabilize training. Our output decoder aggregates temporal spikes to encourage earlier firing, thereby reducing latency. The re-evaluation of the pooling layer demonstrates that max-pooling violates single-spike constraints, which should be avoided, whereas average-pooling preserves them. Experiments show that our framework stabilizes and accelerates training, reduces latency, and achieves state-of-the-art accuracy for step-by-step TTFS SNNs on MNIST (99.48%), Fashion-MNIST (92.90%), CIFAR10 (90.56%), CIFAR100 (70.27%) and DVS Gesture (95.83%). Code and experimental logs are available in: https://github.com/CheKaiWei/ETTFS.

## 1. Introduction

Spiking Neural Networks (SNNs) are recognized as the third generation of neural models characterized by event-driven spike communication (Maass, 1997). This paradigm enables highly energy-efficient computation as validated by neuromorphic hardware including TrueNorth (Merolla et al., 2014), Loihi (Davies et al., 2021), and Tianjic (Pei et al., 2019). Although their biological plausibility benefits neuroscience research (Eliasmith et al., 2012; Stimberg et al., 2019), traditional training methods like Hebbian learning (Hebb, 2005) or Spike-Timing-Dependent Plasticity (STDP) (Bi & Poo, 1998) often yields suboptimal accuracy. To address this, recent advancements integrate deep learning techniques to significantly enhance performance (Fang et al., 2021; Che et al., 2024), establishing SNNs as a critical bridge between neuroscience and computational science.

Neural coding, which is fundamental to SNNs, governs how information is represented through spikes. Among the existing coding methods, Time-to-First-Spike (TTFS) coding encodes information through the timing of the first spike, enabling rapid biological processing (Gollisch & Meister, 2008). TTFS thus provides extreme sparsity and energy efficiency for neuromorphic hardware, attracting significant research interest and yielding methods such as ANN-to-SNN conversion (Rueckauer & Liu, 2018) and direct training (Kheradpisheh et al., 2022). However, strict spike constraints in TTFS SNNs pose significant training challenges. These constraints typically result in substantially lower accuracy than that of traditional SNNs. To address this limitation, we propose an efficient framework that integrates parameter initialization, weight normalization, and a temporal decoder. Our contributions are as follows:

1. We identify the signal diminishing problem caused by Kaiming initialization (Figure 1a) and solve it by proposing a novel parameter initialization method (**ETTFS-init**). This method stabilizes the mean and variance of post-synaptic currents across layers, resulting in a stable distribution as shown in Figure 1b and Section 4.2.

2. We introduce **weight normalization** to stabilize weights around their initialized state, which prevents distribution shifts during training. This approach ac-

---
[1]Shenzhen Graduate School, Peking University, China [2]Pengcheng Laboratory, China [3]School of Computer Science, Peking University, China [4]Shenzhen Institute of Advanced Technology, Chinese Academy of Sciences, China. Correspondence to: Zhengyu Ma <mazhy@pcl.ac.cn>.

*Proceedings of the 43rd International Conference on Machine Learning*, Seoul, South Korea. PMLR 306, 2026. Copyright 2026 by the author(s).

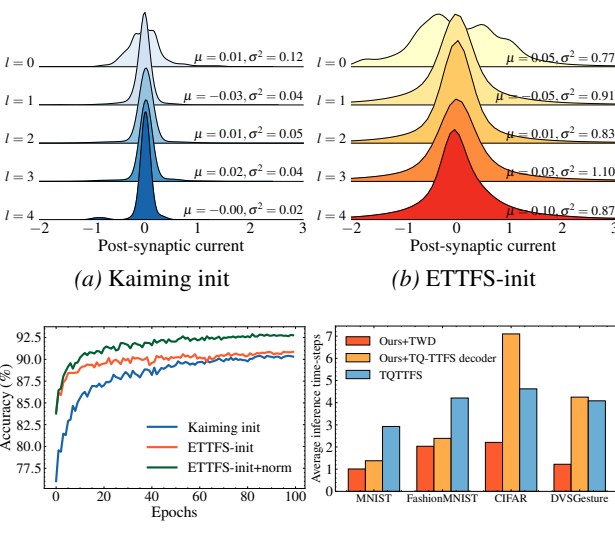

*(a)* Kaiming init     *(b)* ETTFS-init

*(c)* Training accuracy     *(d)* Average inference time-steps

*Figure 1.* An overview of the impacts and performance of the proposed methods. (a) The default Kaiming initialization causes the signal diminishing problem, where the post-synaptic current variance ($\sigma^2$) rapidly decreases across layers ($l$). (b) Our proposed ETTFS-init method regulates these distributions. (c) ETTFS-init accelerates convergence and improves accuracy, with performance further enhanced by weight normalization. (d) Our proposed decoder reduces the average inference time-steps compared to the previous TQ-TTFS decoder across four datasets.

celerates convergence, improves accuracy, and can be fused with synaptic weights without increasing computational overhead during inference, as shown in Figure 1c and Section 4.3.

3. We propose a novel **temporal weighting decoder** to encourage early firing, which improves performance and dramatically decreases the number of inference time-steps, as shown in Figure 1d and Section 4.4.

4. We re-evaluate the pooling layers in TTFS SNNs and argue for using average-pooling to maintain the one-spike characteristic. This is a precondition for the parameter initialization and weight normalization methods described in Section 4.5.

5. Equipped with this proposed efficient training framework, we achieve state-of-the-art accuracy among all step-by-step TTFS SNNs on several benchmark datasets, including MNIST (99.48%), Fashion-MNIST (92.90%), CIFAR10 (90.56%), CIFAR100 (70.29%), and DVS Gesture (95.83%).

## 2. Related Work

### 2.1. Neural Coding in SNNs

Neural coding in SNNs represents information through spike patterns. Rate coding measures average spike counts proportional to input strength, enabling ANN-to-SNN conversion but requiring long time windows (Kim et al., 2018). Consequently, temporal coding using the temporal information of spikes has been explored to reduce the number of time-steps. Among temporal coding methods, phase coding represents values in powers of two (Wang et al., 2022), while burst coding employs graded spikes to surpass rate-based conversion accuracy (Li & Zeng, 2022). TTFS coding represents information through the timing of the first spike. This scheme achieves theoretically minimal energy consumption in SNNs. Consequently, it is a prominent approach for both ANN-to-SNN conversion (Hu et al., 2023) and direct training (Yang et al., 2024).

### 2.2. Initialization of Networks

Proper weight initialization ensures stable information flow in deep neural networks. Similar to recurrent neural networks (RNNs), SNNs suffer from vanishing and exploding gradients due to combined spatial and temporal dimensions (Fang et al., 2021). Consequently, SNN-specific initialization strategies remain significantly underdeveloped.

Existing approaches predominantly adopt generic methods such as Kaiming initialization (He et al., 2015). Ding et al. (Ding et al., 2022) first derived the approximate response curve of spiking neurons, based on which the parameters were initialized to overcome the gradient vanishing problem. However, their method fails to scale to deep SNNs trained on complex datasets. Rossbroich et al. (Rossbroich et al., 2022) proposed an initialization strategy for the Leaky-Integrate-and-Fire (LIF) neuron with exponential current-based synapses but could not extend their method to stateless synapses, which are predominant in deep SNNs. This gap arises because spiking neuron dynamics exhibit greater temporal complexity than the non-linear activations in ANNs.

## 3. Preliminary

### 3.1. Vanilla Spiking Neuron Model

We first introduce the discrete-time spiking neuron model underlying conventional SNNs (Fang et al., 2023a):

$$H[t] = f(V[t-1], X[t]), \qquad (1)$$

$$S[t] = \Theta\left(H[t] - V_{th}\right), \qquad (2)$$

$$V[t] = \begin{cases} H[t] \cdot (1 - S[t]) + V_{reset} \cdot S[t], & \text{hard reset,} \\ H[t] - V_{th} \cdot S[t], & \text{soft reset,} \end{cases} \qquad (3)$$

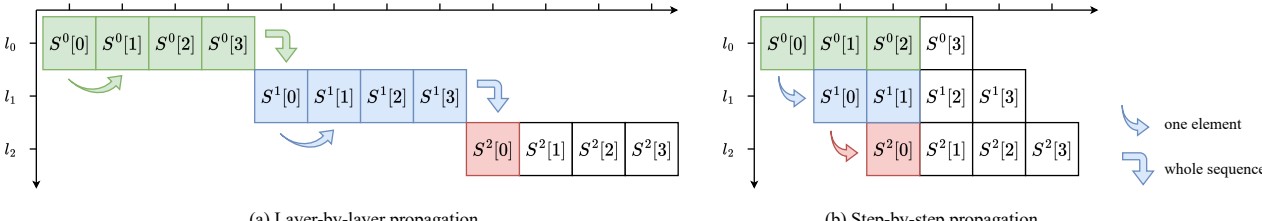

*Figure 2.* Physical latency comparison between (a) layer-by-layer propagation and (b) step-by-step propagation in a three-layer SNN with $T = 4$ time-steps. Each layer consumes one time unit to process one time-step, and the output spike $S^i[t]$ is transmitted immediately after it is generated, where $i$ denotes the layer index and $t$ denotes the time-step. In layer-by-layer propagation, each layer must wait for the complete input spike sequence before producing its own output sequence, which leads to latency accumulation with network depth. In step-by-step propagation, spikes are forwarded across layers at every time-step, which reduces physical latency.

This model updates the neuronal state through three stages: *charge*, *fire*, and *reset*. In Eq. (1), $X[t]$ denotes the input current at time-step $t$, $V[t-1]$ and $V[t]$ denote the membrane potentials before and after the update, respectively, $H[t]$ is the membrane potential after charging but before reset, and $f(\cdot)$ is the neuron-specific charging function. Eq. (2) generates the output spike $S[t]$ when $H[t]$ reaches the threshold $V_{th}$, where $\Theta(\cdot)$ is the Heaviside step function. Eq. (3) then updates the post-spike membrane potential: hard reset clamps the neuron to $V_{reset}$ after firing, whereas soft reset subtracts the threshold $V_{th}$ from $H[t]$.

### 3.2. Propagation Paradigms of TTFS SNNs

TTFS SNNs can be divided into non-causal and causal models according to their temporal dependency. Non-causal models determine output spikes from the complete input spike sequence and therefore restrictly require layer-by-layer execution. ANN-to-SNN conversion is a representative example, because each layer must receive all $T$ time-steps before producing its output sequence.

Causal TTFS models, by contrast, update neuronal states using only the current and past inputs. This property allows two mathematically equivalent execution modes: layer-by-layer propagation for efficient GPU training, and step-by-step propagation for low-latency neuromorphic hardware deployment. As shown in Figure 2, layer-by-layer propagation incurs depth-dependent latency because deeper layers must wait for the full output sequence of shallower layers, whereas step-by-step propagation forwards spikes immediately at each time-step.

In general, non-causal TTFS models achieve higher accuracy because they can optimize with access to the complete input spike sequence, whereas causal models rely only on the current and past inputs at each time-step, which limits their expressive capacity. Our method belongs to the causal paradigm and addresses its main challenge: maintaining the low physical latency of step-by-step propagation without sacrificing accuracy.

## 4. Methods

### 4.1. At-Most-One-Spike Spiking Neuron Model

The At-Most-One-Spike (AMOS) spiking neuron model (Yang et al., 2024) is an extension of the general discrete-time spiking neuron model (Fang et al., 2023b) with the restriction of firing no more than once. The neuronal dynamics of the AMOS spiking neuron are governed by three sequential stages: *charge*, *fire*, and *mask update*, as shown in Figure 3 and the following equations:

$$H[t] = f(H[t-1], X[t]), \quad (4)$$
$$S[t] = (1 - F[t-1]) \cdot \Theta(H[t] - V_{\text{th}}), \quad (5)$$
$$F[t] = F[t-1] + S[t], \quad (6)$$

where $H[t]$ is the membrane potential, $X[t]$ is the input current, $S[t]$ is the output spike, and $F[t]$ is the firing mask at time-step $t$. The neuronal charge function $f$ in Eq. (4) is neuron-specific. For example, the formulation for the IF neuron is:

$$H[t] = H[t-1] + X[t]. \quad (7)$$

The Heaviside function $\Theta(x)$ satisfies $\Theta(x) = 1$ if $x \geq 0$ and $0$ otherwise. Spiking is governed by the mask $F[t]$ that accumulates via Eq. (6), with initial states $H[-1] = F[-1] = 0$. When $S[i] = 1$ at time-step $i$, $F[i] = 1$ prevents subsequent firing. Hence, the neuron can fire no more than one spike and is named the AMOS spiking neuron.

Compared to traditional spiking neurons, the neuronal dynamics of the AMOS neuron omit the reset mechanism, allowing for post-spike resting that could enhance energy efficiency in asynchronous neuromorphic chips. During training, the neuron is forced to fire a spike at the last time-step $t = T - 1$ if it has not fired from $t = 0$ to $t = T - 2$. This setting is also implemented without an if-else statement by updating $S[T-1]$ as:

$$S[T-1] \leftarrow S[T-1] + 1 - F[T-1]. \quad (8)$$

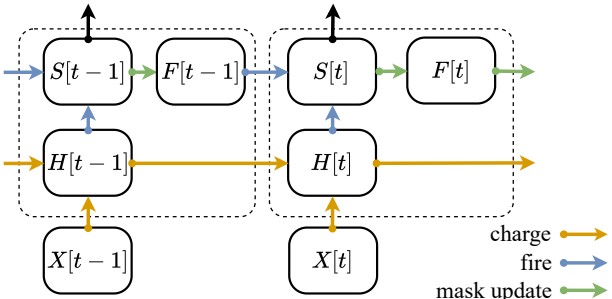

*Figure 3.* The neuronal dynamics of the AMOS neuron include three stages: *charge*, *fire*, and *mask update*. In the *charge* stage, the membrane potential $H[t]$ accumulates via the input $X[t]$. During the *fire* stage, the neuron generates a spike $S[t] = 1$ if both the membrane potential $H[t]$ surpasses the threshold $V_{th}$ and the mask $F[t-1]$ is zero; otherwise, $S[t] = 0$. The mask $F[t]$ is updated by accumulating $S[t]$ at the *mask update* stage.

Eq. (8) is used after Eq. (6) at the last time-step. This setting preserves the one-spike characteristic, which is crucial for theoretical analysis and will be frequently referred to in the following sections. During inference, the AMOS neuron is not forced to fire at the last time-step. Instead, it runs in step-by-step mode. This enables early stopping when a spike occurs in the last layer.

### 4.2. Initialization for TTFS SNNs

The goal of Xavier (Glorot & Bengio, 2010) and Kaiming initialization (He et al., 2015) is to maintain element-wise stability. Specifically, Kaiming initialization aims for the expectation and variance of each element of the pre-activations to be 0 and 1, respectively. Consequently, we also enforce that the initial expectation and variance of the AMOS neuron are maintained at 0 and 1. Without loss of generality, suppose we have an SNN composed of $L$ stacked fully connected AMOS spiking neuron layers. The $l$-th layer can be formulated as:

$$X^l = S^{l-1} W^l, \tag{9}$$
$$S^l = \text{AMOS}(X^l). \tag{10}$$

Here, $S^{l-1} \in \{0, 1\}^{T \times N_l}$ is the input spike from the $(l-1)$-th layer, where $T$ is the number of time-steps and $N_l$ is the number of input features of the $l$-th layer. $W^l \in \mathbb{R}^{N_l \times M_l}$ is the weight of the fully connected synapses, where $M_l$ is the number of output features. $X^l \in \mathbb{R}^{T \times M_l}$ is the input current for the AMOS spiking neurons. $S^l \in \{0, 1\}^{T \times M_l}$ is the output spike of the $l$-th layer, and AMOS represents the AMOS spiking neuron. $S^{l-1}$ maintains the one-spike characteristic: $\sum_{t=0}^{T-1} S^{l-1}[t][j] = 1$ for any $j$-th neuron.

For $X^l[t][i]$, the $i$-th element of the input current of the $l$-th layer at time-step $t$, it is calculated as:

$$X^l[t][i] = \sum_{j=0}^{N_l-1} S^{l-1}[t][j] \cdot W^l[j][i], \tag{11}$$

where $W^l[j][i]$ denotes the synapse weight from $j$-th neuron in $(l-1)$-th layer to $i$-th neuron in $l$-th layer.

Let the weights of the $l$-th layer follow a distribution with mean $\mu_l$ and variance $\sigma_l^2$. Note that $W^l$ and $S^{l-1}$ are independent. Denote the expectation operator as $\mathbb{E}$ and the variance operator as $\mathbb{D}$. Then the expectation $\mathbb{E}(X^l[t][i])$ is:

$$
\begin{aligned}
\mathbb{E}(X^l[t][i]) &= \mathbb{E}(\sum_{j=0}^{N_l-1} S^{l-1}[t][j] \cdot W^l[j][i]) \\
&= \sum_{j=0}^{N_l-1} \mathbb{E}(S^{l-1}[t][j] \cdot W^l[j][i]) \\
&= \sum_{j=0}^{N_l-1} \mathbb{E}(S^{l-1}[t][j]) \cdot \mathbb{E}(W^l[j][i]) \\
&= \sum_{j=0}^{N_l-1} \mathbb{E}(S^{l-1}[t][j]) \cdot \mu_l. \tag{12}
\end{aligned}
$$

Similar to Xavier (Glorot & Bengio, 2010) and Kaiming initialization (He et al., 2015), we also set $\mu_l = 0$ to make $\mathbb{E}(X^l[t][i]) = 0$ in Eq. (12). The variance $\mathbb{D}(X^l[t][i])$ is:

$$
\begin{aligned}
\mathbb{D}(X^l[t][i]) &= \mathbb{D}\left(\sum_{j=0}^{N_l-1} S^{l-1}[t][j] \cdot W^l[j][i]\right) \\
&= N_l \cdot \mathbb{D}(S^{l-1}[t][j] \cdot W^l[j][i]) \\
&= N_l \cdot \left(\mathbb{E}((S^{l-1}[t][j] \cdot W^l[j][i])^2) \right. \\
&\quad \left. - \mathbb{E}^2(S^{l-1}[t][j] \cdot W^l[j][i])\right) \\
&= N_l \cdot (\mathbb{E}((S^{l-1}[t][j])^2) \cdot \mathbb{E}((W^l[j][i])^2) \\
&\quad - \mathbb{E}^2(S^{l-1}[t][j]) \cdot 0) \\
&= N_l \sigma_l^2 \cdot \mathbb{E}((S^{l-1}[t][j])^2) \\
&= N_l \sigma_l^2 \cdot \mathbb{E}(S^{l-1}[t][j]). \tag{13}
\end{aligned}
$$

Unfortunately, Eq. (13) depends on $S^{l-1}[t][j]$, which is not independent and identically distributed, making it impossible to keep $\mathbb{D}(X^l[t][i])$ as a constant. Alternatively, we stabilize the average input current of the neuron for each time step, as follows:

$$\frac{1}{T}\sum_{t=0}^{T-1}\mathbb{D}(X^l[t][i]) = \frac{1}{T}\sum_{t=0}^{T-1}N_l\sigma_l^2 \cdot \mathbb{E}(S^{l-1}[t][j])$$

$$= \frac{N_l\sigma_l^2}{T}\cdot\mathbb{E}(\sum_{t=0}^{T-1}S^{l-1}[t][j])$$

$$= \frac{N_l\sigma_l^2}{T}. \tag{14}$$

When using Kaiming initialization (He et al., 2015), the default weight initialization in PyTorch, the weight will be sampled from a uniform distribution $\mathcal{U}\left(-\frac{1}{\sqrt{N_l}}, \frac{1}{\sqrt{N_l}}\right)$. It satisfies the zero mean of weights according to Eq. (12). However, according to Eq. (14), the variance is:

$$\frac{N_l}{T}\cdot\frac{1}{12}\cdot\left(\frac{1}{\sqrt{N_l}} - \left(-\frac{1}{\sqrt{N_l}}\right)\right)^2 = \frac{1}{3T}, \tag{15}$$

which decays asymptotically to zero as the number of time-steps $T$ increases. As the surrogate function produces near-zero values for membrane potentials far from the threshold, extremely small weights cause the silent neuron problem and vanishing gradients. To maintain initialized weights within a stable range decoupled from $T$, we set $\frac{1}{T}\sum_{t=0}^{T-1}\mathbb{D}(X^l[t][i]) = 1$. From Eq. (14), $\sigma_l^2$ must satisfy $\sigma_l^2 = \frac{T}{N_l}$. We thus propose ETTFS-init: for layer $l$ in a TTFS SNN ($T$ time-steps), initialize $W^l$ from the following uniform distribution:

$$W^l \sim \mathcal{U}\left(-\sqrt{\frac{3T}{N_l}}, \sqrt{\frac{3T}{N_l}}\right). \tag{16}$$

For fully-connected layers, $N_l$ denotes the number of input features. For convolutional layers, $N_l = C_{in}^l \cdot N_{kernel}^l$, where $C_{in}^l$ is the number of input channels and $N_{kernel}^l$ is the number of elements in the kernel. With Eq. (16), the input current for the $l$-th AMOS spiking neuron layer is stabilized:

$$\mathbb{E}(X^l[t][i]) = 0, \quad \frac{1}{T}\sum_{t=0}^{T-1}\mathbb{D}(X^l[t][i]) = 1. \tag{17}$$

### 4.3. Weight Normalization

Although the ETTFS-init method offers a reasonable starting point, the distribution of weights may shift during gradient descent and consequently violate Eq. (17) during the training process as shown in Figure 4.

To address this, we employ weight normalization (Qiao et al., 2019) to maintain a consistent weight distribution:

$$W^l \leftarrow \frac{W^l - \mathbb{E}(W^l)}{\sqrt{\mathbb{D}(W^l) + \epsilon}}\cdot\sigma_{W^l}, \tag{18}$$

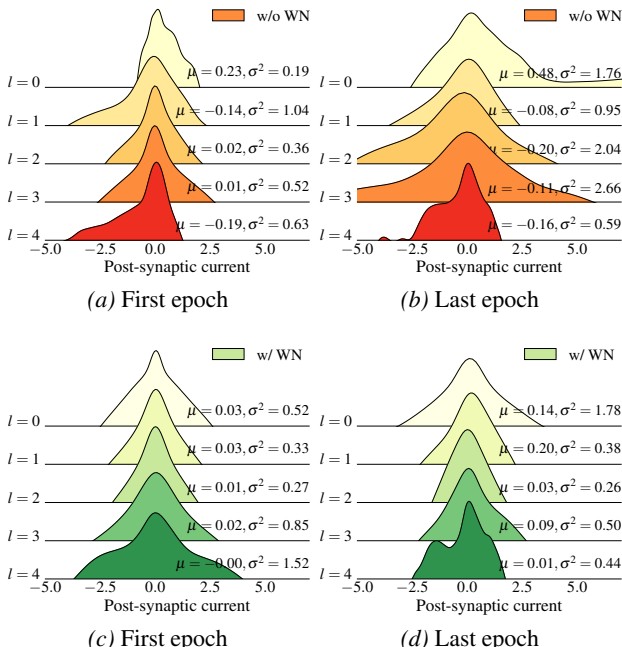

*Figure 4.* Comparison of the distribution of $X^l$ during training with the ETTFS-init method only and with an additional weight normalization (WN) method. Without normalization, the distribution of $X^l$ shifts significantly as the epoch increases. With normalization, the distribution of $X^l$ remains stable and shifts only minimally.

where $W^l$ is the weight of the $l$-th layer, $\mathbb{E}(W^l)$ and $\mathbb{D}(W^l)$ denote the mean and variance of $W^l$, respectively; $\epsilon$ is a small constant for numerical stability; and $\sigma_{W^l}$ is the standard deviation of the initial weights determined by Eq. (16) when using ETTFS-init.

Eq. (18) constrains weight scaling and limits representational capacity. Inspired by batch normalization (Ioffe & Szegedy, 2015) and layer normalization (Lei Ba et al., 2016), which resolve fixed-scale issues via learnable affine transforms, we apply a learnable affine transform to the synaptic output $X^l$ with weight $\gamma^l$ and bias $\beta^l$:

$$X^l \leftarrow \gamma^l X^l + \beta^l, \tag{19}$$

where $\gamma^l$ and $\beta^l$ are initialized as all-ones and all-zeros tensors, respectively. Both weight normalization and the learnable affine transform are applied only during training; at inference, the affine transform is absorbed into the synaptic weights. Consequently, the fused weight $\tilde{W}^l$ and bias $\tilde{B}^l$ for the $l$-th layer are given by:

$$\tilde{W}^l = \gamma^l \cdot W^l, \quad \tilde{B}^l = \beta^l. \tag{20}$$

Fusing the affine transform into synaptic weights eliminates computational overhead during inference. During training, weight normalization prevents distributional shifts and maintains a stable activation distribution for $X^l$. Thus, the model achieves both inference efficiency and training stability.

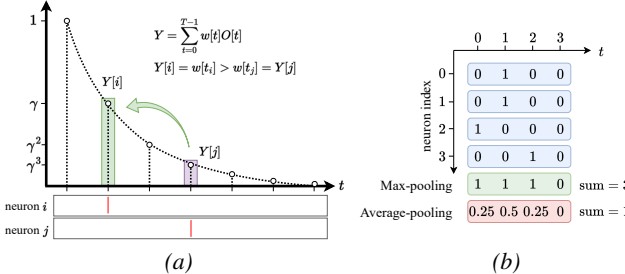

*Figure 5.* (a) The mechanism of the temporal weighting decoder. The weight $w[t]$ is set as an exponential or linear decay function of $t$, ensuring that neurons firing earlier produce larger outputs (e.g., $t_i < t_j$ implies $Y[i] > Y[j]$). (b) Comparison between average-pooling and max-pooling in TTFS SNNs.

## 4.4. Temporal Weighting Decoder

Traditional SNN decoders use output layer firing rates (Wu et al., 2018), which are ineffective for TTFS networks due to uniform firing rates ($1/T$) across neurons during training. The temporal nature of TTFS coding suggests decoding through the earliest spike timing. More specifically, suppose the SNN is designed for a classification task with $C$ classes and the number of time-steps is $T$. The output of the SNN is the tensor $O \in \{0, 1\}^{T \times C}$. The predicted class is then:

$$Y_{\text{predict}} = \text{argmin}_i\{t \mid O[t][i] = 1\}. \quad (21)$$

Equation (21) is used in inference, not training. However, since argmin is non-differentiable, this rule is used only during inference—not training. Therefore, an effective decoder approach for TTFS SNNs must have two key features: differentiability and an emphasis on early firing.

Existing TTFS decoders such as TQ-TTFS decoder exhibit computational complexity and underutilize temporal information (Yang et al., 2024). To solve these issues, we propose the temporal weighting decoder, which aggregates the outputs using decayed temporal weights $w \in \mathbb{R}^T$ over time-steps, prioritizing early firing while preserving differentiability as shown in Figure 5a. Consequently, the decoded output $Y \in \mathbb{R}^C$ is defined as the weighted sum of outputs over time-steps:

$$Y = \sum_{t=0}^{T-1} w[t] \cdot O[t]. \quad (22)$$

We set $w[0] = 1$ and $w[t] > w[t+1]$ for any $t \in \{0, 1, \ldots, T-2\}$. The weighting factor $w$ can be set as an exponential decay function or a linear decay function:

$$w[t] = \gamma^{-t}, \quad w[t] = \gamma \cdot \frac{T-t}{T}. \quad (23)$$

This formulation ensures differentiability prioritizes earlier firing times via scaling factor $\gamma > 1$. Thus, if neuron $i$ fires at $t_i$ and neuron $j$ fires at $t_j$ with $t_i < t_j$, the decoder yields a higher contribution from neuron $i$:

$$Y[i] = w[t_i] > w[t_j] = Y[j]. \quad (24)$$

Derived from Eq. (24), the neuron that has the earliest firing time-step also enjoys the largest decoded value:

$$\text{argmin}_i\{t \mid O[t][i] = 1\} = \text{argmax}_i\{Y[i]\}. \quad (25)$$

Most prior TTFS methods use spike timing decoder requiring long time windows for accurate gradients and large computational overhead for converting continuous timing to binary spikes (Wei et al., 2023). In contrast, our temporal weighting decoder achieves lower latency through step-by-step propagation while preserving strictly sparse binary representations.

## 4.5. Pooling for One-Spike Characteristic

Pooling layers reduce spatial dimensions to extract compact features. However, prior work (Yang et al., 2024) adopts max-pooling without considering its compatibility with TTFS coding. Consider a pooling window over $n$ one-spike input sequences $S_{\text{in}} \in \{0, 1\}^{T \times n}$, the sum of the max-pooled output $S_{\text{out}} \in \{0, 1\}^{T \times 1}$ is:

$$\sum_{t=0}^{T-1} S_{\text{out}}[t] = \sum_{t=0}^{T-1} \max(S_{\text{in}}[t][0], S_{\text{in}}[t][1], ..., S_{\text{in}}[t][n-1])$$
$$= \sum_{t=0}^{T-1} \min\left(\sum_{i=0}^{n-1} S_{\text{in}}[t][i], 1\right). \quad (26)$$

Max-pooling selects the maximum across spatial dimension at each time-step. $\sum_{t=0}^{T-1} S_{\text{out}}[t] > 1$ occurs when more than one neuron in the pooling window fires at different time-steps, which seriously violates the one-spike assumption. Conversely, average-pooling does not suffer from this problem as it averages over the spatial dimension $n$ while the sum of each sequence is always 1:

$$\sum_{t=0}^{T-1} S_{\text{out}}[t] = \sum_{t=0}^{T-1} \frac{1}{n} \sum_{i=0}^{n-1} S_{\text{in}}[t][i] = 1. \quad (27)$$

Moreover, as both average-pooling and convolutional layers are linear operations, they can theoretically be fused into a single linear layer to reduce computation overhead.

## 5. Experiments

In this section, we evaluate the accuracy and latency of ET-TFS based on SpikingJelly framework (Fang et al., 2023a).

| Method | MNIST | | Fashion-MNIST | | CIFAR10 | | CIFAR100 | | DVS Gesture | |
|---|---|---|---|---|---|---|---|---|---|---|
| | Model | Acc. | Model | Acc. | Model | Acc. | Model | Acc. | Model | Acc. |
| **Conversion** | | | | | | | | | | |
| TTFS clamped (Rueckauer & Liu, 2018) | LeNet-5 | 98.53% | - | - | - | - | - | - | - | - |
| LC-TTFS (Yang et al., 2023) | - | - | - | - | VGG-16 | **92.72%** | VGG-16 | 70.15% | - | - |
| T2FSNN (Park et al., 2020) | - | - | - | - | VGG-16 | 91.43% | VGG-16 | 68.79% | - | - |
| **Direct training** | | | | | | | | | | |
| Mostafa (Mostafa, 2017) | FC800-FC10 | 97.5% | - | - | - | - | - | - | - | - |
| Comsa et al. (Comsa et al., 2020) | FC340-FC10 | 97.9% | - | - | - | - | - | - | - | - |
| Göltz et al. (Göltz et al., 2021) | FC350-FC10 | 97.2% | - | - | - | - | - | - | - | - |
| S4NN (Kheradpisheh & Masquelier, 2020) | FC600-FC10 | 97.4% | FC1000-FC10 | 88.0% | - | - | - | - | - | - |
| BS4NN (Kheradpisheh et al., 2022) | FC600-FC10 | 97.0% | FC1000-FC10 | 87.3% | - | - | - | - | - | - |
| Zhang et al. (Zhang et al., 2021) | FC400-FC10 | 98.1% | SCNN[1] | 90.1% | - | - | - | - | - | - |
| Sakemi et al. (Sakemi et al., 2023) | SCNN[2] | 99.25% | SCNN[2] | 89.5% | SCNN[3] | 80.0% | - | - | - | - |
| STiDi-BP (Mirsadeghi et al., 2023) | SCNN[4] | 99.2% | SCNN[5] | 92.8% | - | - | - | - | - | - |
| DTA-TTFS (Wei et al., 2023) | SCNN[1] | 99.4% | - | - | VGG16 | 93.55%[P] | VGG16 | 69.66%[P] | - | - |
| | | | | | VGG16 | 80.38%[†] | VGG16 | 1 %[†] | | |
| TQ-TTFS (Yang et al., 2024) | FC400-FC10 | 98.6% | FC400-FC400-FC10 | 88.1% | SCNN[6] | 67.47 %[†] | SCNN[6] | 1%[†] | SCNN[7] | 88.89%[†] |
| | | | SCNN[1] | 90.2% | | | | | | |
| **ETTFS (ours)** | FC400-FC10 | $98.67^{\pm 0.14}$% | FC400-FC400-FC10 | $90.21^{\pm 0.16}$% | SCNN[6] | $93.89^{\pm 0.28}$%[P] | SCNN[6] | $72.70^{\pm 0.44}$%[P] | SCNN[7] | $95.83^{\pm 0.69}$% |
| | SCNN[1] | $99.48^{\pm 0.07}$% | SCNN[5] | $92.90^{\pm 0.11}$% | SCNN[6] | $90.56^{\pm 0.33}$% | SCNN[6] | $70.27^{\pm 0.52}$% | | |

*Table 1.* Comparison on MNIST, Fashion-MNIST, CIFAR10/100, and DVS Gesture datasets. [†] denotes our implementation. [P] denotes using pre-trained ReLU weights. Notation: FCn (fully-connected layer with n output features). Refer to the Supplementary Materials Table 9 for more details about the symbolic representations of the network structure.

## 5.1. Comparison with the State-of-the-Art

We evaluate ETTFS on the static MNIST, Fashion-MNIST, CIFAR10/100, and the neuromorphic DVS Gesture dataset (Amir et al., 2017).

### 5.1.1. ACCURACY

ETTFS achieves state-of-the-art accuracy among all direct training methods on all benchmarks. Both DTA-TTFS and conversion-based methods require a large VGG16 model (138.36M parameters) and significantly more time-steps ($> 100$ steps). While DTA-TTFS employs direct training, it remain reliant on costly ReLU pre-trained weights.

In contrast, ETTFS uses a compact SCNN (70.11M parameters) and outperforms DTA-TTFS both with and without pre-training with only 2.93 average time-steps. For a fair comparison to other methods, we thus report $90.56\%$ as our CIFAR10 baseline. Furthermore, ETTFS's step-by-step propagation matches the asynchronous, event-driven nature of neuromorphic hardware, whereas conversion methods rely on layer-by-layer inference, which incurs high latency.

DVS data requires spikes in different time-steps to represent motion, which conflicts with the strict spike constraint of TTFS coding. Consequently, training TTFS SNNs on DVS datasets is challenging and no prior TTFS results exist in DVS Gesture dataset. To address this, we provide the first implementation of TQ-TTFS and the proposed ETTFS on DVS benchmarks, where competing methods typically fail to converge.

| Method | MNIST | Fashion-MNIST | CIFAR10/100 | DVS Gesture |
|---|---|---|---|---|
| T2FSNN | 20[*] (40) | - | 400[*] (680) | - |
| Mostafa | 3.5[*] (5[*]) | - | - | - |
| Comsa et al. | 20[*] (50[*]) | - | - | - |
| S4NN | 89.7 (256) | - | - | - |
| BS4NN | 112.2[*] (256) | 100.20[*] (256) | - | - |
| STiDi-BP | 71.1 (100) | 61.3 (100) | - | - |
| DTA-TTFS | - | - | 160 (160) | - |
| TQ-TTFS | 2.93[†] (8) | 4.21[†] (8) | 4.62[†] (8) | 4.08[†] (8) |
| **ETTFS (ours)** | **1.01** (8) | **2.03** (8) | **2.93** (8) | **1.20** (8) |

*Table 2.* Comparison of time-steps on MNIST, Fashion-MNIST, CIFAR10/100, and DVS Gesture datasets. Each element denotes: inference time-steps (training time-steps). Some prior works report results in figures (denoted by [*]). [†] denotes our implementation.

| | time-steps | $N_s$ | SOPs | E1 | E2 (Normalized) | E3 |
|---|---|---|---|---|---|---|
| Rate coding | 8 | $4.54 \times 10^5$ | $11.02 \times 10^6$ | 87.79uJ | 1 | 108.59uJ |
| TQ-TTFS | 4.62 | $2.71 \times 10^5$ | $3.78 \times 10^6$ | 20.59uJ | 0.596 | 39.77uJ |
| ETTFS | 2.93 | $2.69 \times 10^5$ | $3.45 \times 10^6$ | 12.19uJ | 0.592 | 27.56uJ |

*Table 3.* Comparison of inference time-steps, the number of spikes $N_s$, synaptic operations (SOPs), and three estimated energy consumption metrics on CIFAR10.

### 5.1.2. LATENCY AND ENERGY CONSUMPTION

TTFS SNNs enable early inference stopping upon the first output spike. This early stopping reduces inference time-steps far below training. Table 2 reports the average time-steps over the test set. Conversion methods require substantially more time-steps and suffer from latency inflation in hardware due to mandatory layer-by-layer propagation. In contrast, direct training methods like TQ-TTFS and ours leverage step-by-step propagation to achieve minimal training time-steps.

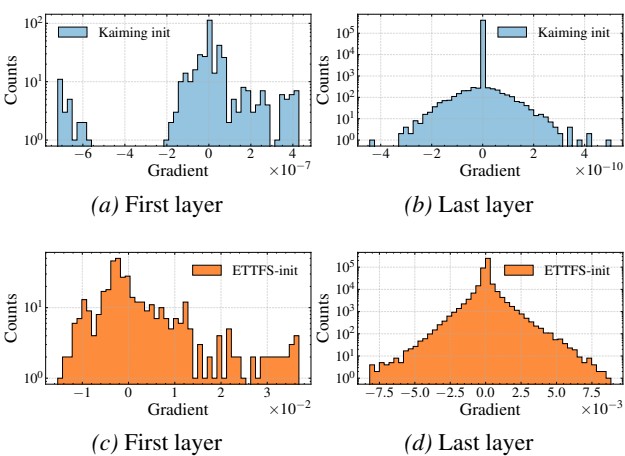

*(a)* First layer      *(b)* Last layer

*(c)* First layer      *(d)* Last layer

*Figure 6.* Comparison of the gradients of weights initialized by the Kaiming initialization and the ETTFS-init method. The Kaiming initialization results in extremely small gradients, with a scale of less than $10^{-7}$. In contrast, our ETTFS-init method leads to appropriate gradients with a scale larger than $10^{-3}$.

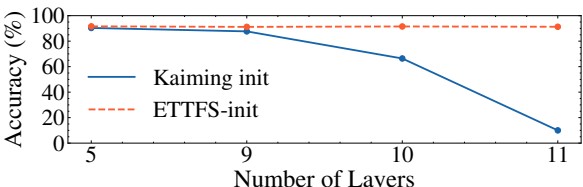

*Figure 7.* Evaluation of the stability of ETTFS-init on Fashion-MNIST. Increasing the number of layers from 5 to 11.

Prior SNN research emphasizes synaptic energy, neglecting memory access costs (Chowdhury et al., 2021). To achieve a comprehensive measurement, we measure energy consumption using three distinct metrics:

**E1. Event-driven synaptic-operation energy.** The first estimate evaluates the energy spent by spike-triggered synaptic accumulations. Following the commonly adopted 45 nm CMOS setting, we use $E_{AC} = 0.9$ pJ for one accumulation operation (Rathi & Roy, 2021). Given an SNN with $L$ layers, the estimated energy is

$$E_1 = E_{AC} \sum_{l=0}^{L-1} T \, \text{FLOPs}(l) \, r_l, \qquad (28)$$

where $T$ denotes the inference time-steps, $\text{FLOPs}(l)$ is the operation count of layer $l$ in its dense counterpart, and $r_l$ is the average spike firing rate of that layer. However, this SOP-based metric overemphasizes synaptic sparsity while neglecting the dominant impact of spike count on neuromorphic hardware energy.

**E2. Spike-count-aware static dynamic energy.** The second estimate metric captures the temporal and spiking energy costs in neuromorphic hardware (Wei et al., 2023). It

is defined as:

$$E_2 = \alpha_{\text{stat}} T + \alpha_{\text{dyn}} N_{\text{s}}, \qquad (29)$$

where $N_{\text{s}}$ is the total spike count over all neurons and time-steps. The coefficients $\alpha_{\text{stat}}$ and $\alpha_{\text{dyn}}$ are hardware-specific constants that characterize static and dynamic energy, respectively. For the TrueNorth platform (Merolla et al., 2014), we set $(\alpha_{\text{stat}}, \alpha_{\text{dyn}}) = (0.6, 0.4)$.

**E3. Memory access energy** metric models the memory access energy of neuromorphic chips in Spikesim (Moitra et al., 2023). We report the results for a 262 M-neuron scenario with input dimensions of $C_i, C_o, k, H, W = 256, 256, 3, 32, 32$.

ETTFS shows advantages over rate coding and TQ-TTFS in terms of time-steps, number of spikes, SOPs, and three different energy consumption metrics due to temporal weighting decoder's early stopping mechanism and temporal decay strategy, as shown in Table 3.

## 5.2. Ablation Study

Our methods achieve high accuracy and low latency compared to prior work. We conduct ablation studies to examine the impact of each component. Further ablation study on time-step and post-synaptic variance impact are in Supplementary Materials A.

### 5.2.1. ETTFS-INIT ON BACKWARD PROPAGATION

We analyze backward propagation gradients through weight gradients $\frac{\partial \mathcal{L}}{\partial W^l}$ in SCNN[5] (Figure 6). With Kaiming initialization, gradients scale from $10^{-7}$ to $10^{-10}$ across layers, leading to the diminishing post-synaptic currents in Figure 1a. In contrast, ETTFS-init maintains stable gradient scales, thereby mitigating the vanishing gradient problem.

### 5.2.2. VERIFICATION ON DEEP TTFS SNNS

Deeper networks suffer from increased training instability especially in TTFS SNNs. For networks expanded from 5 to 11 layers, Kaiming initialization causes accuracy degradation and convergence failure, whereas ETTFS-init maintains high accuracy, as shown in Figure 7.

### 5.2.3. ACCURACY CONTRIBUTIONS

We evaluate each component of ETTFS on Fashion-MNIST as shown in Table 4. The baseline is SCNN[5] with max-pooling and the TQ-TTFS decoder. Starting from the baseline SCNN[5], ETTFS-init accelerates convergence and boosts accuracy compared with Kaiming initialization; adding weight normalization further enhances performance, as shown in Figure 8.

| ETTFS-init | Avg-pooling | Norm | Norm with affine | TWD | Accuracy |
|:---:|:---:|:---:|:---:|:---:|:---:|
| | | | | | 89.61% |
| | | | | ✓ | 89.72% |
| ✓ | | | | | 90.27% |
| ✓ | | | | ✓ | 90.47% |
| | ✓ | | | ✓ | 90.41% |
| ✓ | ✓ | | | ✓ | 90.88% |
| ✓ | ✓ | ✓ | | ✓ | 91.61% |
| ✓ | ✓ | | ✓ | ✓ | **92.90%** |

*Table 4.* Evaluating the components of ETTFS on the Fashion-MNIST dataset. TWD denotes the temporal weighting decoder. The baseline uses max-pooling with the TQ-TTFS decoder.

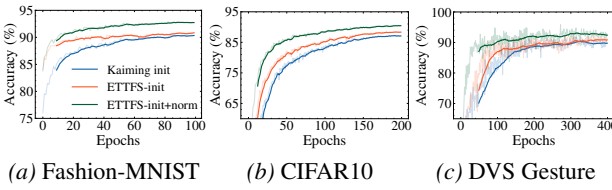

*(a)* Fashion-MNIST    *(b)* CIFAR10    *(c)* DVS Gesture

*Figure 8.* Comparison of the epoch-accuracy curves of the Kaiming and ETTFS initialization methods on three datasets. The curves in dark colors represent the moving average of the test accuracy, while the original data are plotted in light colors.

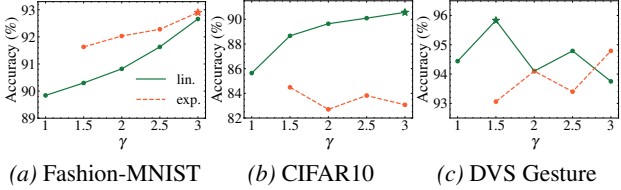

*(a)* Fashion-MNIST    *(b)* CIFAR10    *(c)* DVS Gesture

*Figure 9.* Comparisons of weight decay methods with different $\gamma$ values for the temporal weighting decoder method on three datasets. The highest accuracy on each dataset is marked by ★.

### 5.2.4. WEIGHT DECAY STRATEGIES FOR TEMPORAL WEIGHTING DECODER

Larger $\gamma$ values enhance early-spike emphasis but risk vanishing gradients. We evaluate exponential and linear decay methods across three datasets, observing an optimal $\gamma = 3$ for Fashion-MNIST (exponential decay) and CIFAR10 (linear decay), while DVS Gesture shows stable accuracy for both methods, as shown in Figure 9. Consequently, $\gamma$ requires dataset-specific empirical tuning.

## 6. Conclusion

We propose ETTFS, a training framework for high performance TTFS SNNs. It integrates weight initialization, normalization, temporal decoding, and average-pooling to stabilize training dynamics and reduce inference latency. Consequently, it achieves state-of-the-art accuracy among directly trained TTFS SNNs. Currently, our paper only focuses on small-scale datasets, as do most TTFS SNNs, rather than large-scale datasets such as ImageNet. Therefore, future research should address these limitations to bridge the performance gap and broaden their applicability.

## Acknowledgements

We sincerely apologize for the omission of Timothée Masquelier, who should have been listed as the 7-th author. He was inadvertently omitted during the OpenReview submission process due to a submission oversight on our part. After the submission was completed, the author list could no longer be modified at this stage due to ICML policy constraints. We are deeply grateful for Timothée Masquelier's contributions to and support of this research. Readers are encouraged to cite the arXiv version of this work: https://arxiv.org/abs/2410.23619.

The study was funded by the Guangdong S&T Program (2024B0101010003), the National Natural Science Foundation of China under contracts No. 62425101, No. 62332002, No.62088102, the major key project of the Peng Cheng Laboratory (PCL2025A02), and Shenzhen KQTD (No.20240729102051063).

## Impact Statement

This paper presents work whose goal is to advance the field of machine learning. There are many potential societal consequences of our work, none of which we feel must be specifically highlighted here.

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

## A. Further Ablation Study

### A.1. Time-step impact

We applied a fixed 8 time-steps to ensure a fair comparison with prior work. To further assess the effect of time-step variation, we evaluated configurations from 2 to 12 time-steps on three key datasets; results are reported in Table 5.

| T | Fashion-MNIST (%) | DVSGesture (%) | CIFAR10 (%) |
|---|---|---|---|
| 2 | 87.02 | 71.18 | 88.97 |
| 3 | 87.07 | 96.53 | 90.82 |
| 4 | 89.63 | 95.83 | 91.21 |
| 5 | 91.86 | 95.49 | 90.58 |
| 6 | 92.53 | **95.86** | **90.72** |
| 7 | 92.85 | 94.44 | 90.22 |
| 8 | 92.90 | 95.83 | 90.56 |
| 9 | 92.87 | 93.06 | 90.12 |
| 10 | **93.01** | 94.12 | 89.84 |
| 11 | 92.75 | 93.40 | 89.69 |
| 12 | 92.75 | 92.36 | 89.68 |

*Table 5.* Time-step impact on three datasets.

### A.2. Post-synaptic variance impact

According to Eq. (14), the average of input current variance at each time-step given by:

$$\frac{1}{T} \sum_{t=0}^{T-1} \mathbb{D}(X^l[t][i]) = \frac{N_l \sigma_l^2}{T}. \tag{30}$$

When using Kaiming initialization, the default weight initialization in PyTorch, the weight will be sampled from a uniform distribution $\mathcal{U}\left(-\frac{1}{\sqrt{N_l}}, \frac{1}{\sqrt{N_l}}\right)$. The variance becomes:

$$\frac{N_l}{T} \cdot \frac{1}{12} \cdot (\frac{1}{\sqrt{N_l}} - (-\frac{1}{\sqrt{N_l}}))^2 = \frac{1}{3T}, \tag{31}$$

As $T$ increases, the variance approaches:

$$T \to \infty, \quad \frac{1}{T} \sum_{t=0}^{T-1} \mathbb{D}(X^l[t][i]) \to 0 \tag{32}$$

Many TTFS SNNs require large time-steps (e.g., 100), resulting in $\frac{1}{T} \sum_{t=0}^{T-1} \mathbb{D}(X^l[t][i]) < 0.003$. It is a extremely small value, which will cause the silence problem during forward propagation, leading to vanishing gradients as the surrogate function produces near-zero values when membrane potentials are distant from the threshold.

Therefore, we set the value to 1 as an normal choice. The variance can also be set to other values, as long as it can

decouple $T$ and will not lead to over-small weights. But our analysis across datasets confirms that a variance of 1 consistently outperforms smaller variances, which often lead to suboptimal performance as shown in Table 6.

| $\sigma_l^2$ | Fashion-MNIST (%) | DVSGesture (%) | CIFAR10 (%) |
|---|---|---|---|
| 4 | 92.31 | 95.14 | 90.45 |
| 2 | 92.85 | 94.80 | 90.48 |
| 1 | **92.9** | **95.83** | **90.56** |
| 0.5 | 93.05 | 93.75 | 90.02 |
| 0.25 | 91.83 | 92.08 | 88.64 |

*Table 6.* Comparison with difference initial values of varience across different datasets.

### A.3. Comparison with rate coding

To facilitate a clearer comparison of the performance gap with rate coding with the same network structure and trained it on the same datasets. The results are reported in Table 7.

## B. Experimental Settings

To ensure reproducibility, we provide all experimental settings in Table 8. Additionally, all source code, trained model weights, and training logs for each benchmark are included in the Supplementary Material, enabling full replication of our results. All experiments are conducted on a single NVIDIA RTX 4090 GPU with 24GB memory. For a fair comparison, we use the same network structures as previous methods. The detailed layer descriptions and output sizes can be found in Table 9. Code and experimental logs are available in: `https://github.com/CheKaiWei/ETTFS`.

| Method | MNIST | | Fashion-MNIST | | CIFAR10 | | CIFAR100 | | DVS Gesture | |
|---|---|---|---|---|---|---|---|---|---|---|
| | Model | Acc. | Model | Acc. | Model | Acc. | Model | Acc. | Model | Acc. |
| Rate | SCNN[1] | **99.54%** | SCNN[5] | **93.21%** | SCNN[6] | **94.29%** | SCNN[6] | **73.21%** | SCNN[7] | 93.47% |
| **ETTFS (ours)** | SCNN[1] | 99.48% | SCNN[5] | 92.90% | SCNN[6] | 93.89% | SCNN[6] | 72.70% | SCNN[7] | **95.83%** |

*Table 7.* Comparison between rate coding and TTFS coding on MNIST, Fashion-MNIST, CIFAR10, CIFAR100, and DVS Gesture.

| Dataset | **Fashion-MNIST & MNIST** | **CIFAR10** | **DVS Gesture** |
|---|---|---|---|
| Batch size | 128 | 64 | 16 |
| Epochs | 100 | 200 | 400 |
| Optimizer | AdamW | SGD | AdamW |
| LR | $1e-3$ | $1e-6 \rightarrow 0.1$ | $1e-6 \rightarrow 0.1$ |
| Loss function | MSE | CrossEntropy | MSE |
| Time step | 8 | 8 | 8 |
| DA | Mixup | Mixup, cutmix, horizontal flip, autoaugment, random erase | Random temporal delete |

*Table 8.* Experimental settings on MNIST, Fashion-MNIST, CIFAR10, and DVS Gesture datasets. LR denotes learning rate, DA denotes data augmentation.

| # | Layer Description |
|---|---|
| **FC400-FC10 (MNIST)** | |
| FC400 | lin. $28 \times 28, 400$ |
| FC10 | lin. $400, 10$ |
| TWD | temporal weighting decoder |
| **FC400-FC400-FC10 (Fashion-MNIST)** | |
| FC400 | lin. $28 \times 28, 400$ |
| FC400 | lin. $400, 400$ |
| FC10 | lin. $400, 10$ |
| TWD | temporal weighting decoder |
| **SCNN[1] (MNIST)** | |
| C16K5 | conv. $1 \times 5 \times 5 \times 16$ stride 1 |
| P2 | avg. $2 \times 2$ |
| C32K5 | conv. $16 \times 5 \times 5 \times 32$ stride 1 |
| P2 | avg. $2 \times 2$ |
| FC800 | lin. $512, 800$ |
| FC128 | lin. $800, 128$ |
| FC10 | lin. $800, 10$ |
| TWD | temporal weighting decoder |
| **SCNN[5] (Fashion-MNIST)** | |
| C20K5 | conv. $1 \times 5 \times 5 \times 20$ stride 1 |
| P2 | avg. $2 \times 2$ |
| C40K5 | conv. $20 \times 5 \times 5 \times 40$ stride 1 |
| P2 | avg. $2 \times 2$ |
| FC1000 | lin. $640, 1000$ |
| FC10 | lin. $1000, 10$ |
| TWD | temporal weighting decoder |

| # | Layer Description |
|---|---|
| **SCNN[6] (CIFAR10/100)** | |
| C256K3*3 | conv. $3 \times 3 \times 3 \times 256$ stride 1 *3 |
| P2 | avg. $2 \times 2$ |
| C256K3*3 | conv. $256 \times 3 \times 3 \times 256$ stride 1 *3 |
| P2 | avg. $2 \times 2$ |
| FC4096 | lin. $16384, 4096$ |
| FC10/100 | lin. $4096, 10/100$ |
| TWD | temporal weighting decoder |
| **SCNN[7] (DVS Gesture)** | |
| C128K3 | conv. $2 \times 3 \times 3 \times 128$ stride 1 |
| P2 | avg. $2 \times 2$ |
| C128K3 | conv. $2 \times 3 \times 3 \times 128$ stride 1 |
| P2 | avg. $2 \times 2$ |
| C128K3 | conv. $2 \times 3 \times 3 \times 128$ stride 1 |
| P2 | avg. $2 \times 2$ |
| C128K3 | conv. $2 \times 3 \times 3 \times 128$ stride 1 |
| P2 | avg. $2 \times 2$ |
| C128K3 | conv. $2 \times 3 \times 3 \times 128$ stride 1 |
| P2 | avg. $2 \times 2$ |
| FC521 | lin. $2048, 512$ |
| FC11 | lin. $512, 11$ |
| TWD | temporal weighting decoder |

*Table 9.* Detailed architecture on MNIST, Fashion-MNIST, CIFAR10/100, and DVS Gesture. Notation: FCn (fully-connected layer with n output features), CmKn (convolutional layer with m output channels and n kernel size), Pn (pooling layer with stride n). SCNN[1] is C16K5-P2-C32K5-P2-FC128-FC10. SCNN[2] is C6K5-P2-C16K5-P2-FC400-FC400-10. SCNN[3] is C32K3-P2-C64K3-P2-C128K3-P2-FC10. SCNN[4] is C40K5-P2-FC1000-FC10. SCNN[5] is C20K5-P2-C40K5-P2-FC1000-FC10. SCNN[6] is {{C256K3}*3-P2}*2-FC2048-FC10. SCNN[7] is {C128K3-P2}*5-FC512-FC11.

