# OpenReview forum: "Efficiently Training Time-to-First-Spike Spiking Neural Networks from Scratch"
_ICML.cc/2026/Conference — ICML 2026 spotlight_

### Official Review · Reviewer_1jbn · 2026-03-02

**Soundness:** 3
**Presentation:** 2
**Significance:** 4
**Originality:** 3
**Overall Recommendation:** 5
**Confidence:** 4

**Summary:**

This paper presents an efficient training framework for Time-to-First-Spike (TTFS) spiking neural networks that reduces computational complexity without sacrificing differentiability. The authors firstly introduce the AMOS neuron model to facilitate single-spike transmission, supported by a novel ETTFS-init initialization scheme and a normalization strategy. Together, these techniques stabilize synaptic current distributions across layers, effectively mitigating signal attenuation and distributional shifts. To further enhance performance, a temporal weighting decoding mechanism is employed to prioritize early spikes, maintaining both differentiability and low overhead. Finally, the integration of average-pooling preserves the inherent one-spike sparsity of the TTFS architecture. Experimental results demonstrate that the proposed framework achieves high accuracy across multiple benchmark datasets.

**Compliance With Llm Reviewing Policy:**

Affirmed.

**Final Justification:**

Overall, the paper is well structured and supported by comprehensive experimental results. The authors’ response has satisfactorily addressed my previous concerns.

**Key Questions For Authors:**

1. While average-pooling preserves the temporal sum of 1 (Eq. 24), dividing by the spatial dimension $n$ outputs continuous, fractional values (e.g., 0.25) rather than binary \{0, 1\} spikes. How do subsequent layers process these non-binary inputs without violating strictly AMOS constraint?
2. The authors admit that $\gamma$ requires "dataset-specific empirical tuning," observing that an exponential decay works best for Fashion-MNIST, while linear decay works best for CIFAR10. Is there a theoretical guideline or heuristic to select these parameters for unseen datasets without exhaustive search?
3. AMOS neurons are forced to fire at the last time-step during training, but permitted to remain silent during inference for early stopping. Does this structural mismatch cause a distributional shift in post-synaptic currents during evaluation, and if so, how is it mitigated?
4. Why do prior methods like TQ-TTFS and DTA-TTFS struggle to converge on complex datasets like CIFAR-100 without the aid of pre-trained weights?
If the authors can address my concerns, I would consider raising my score.

**Limitations:**

yes

**Strengths And Weaknesses:**

**Strengths**
1. The authors provide a strong mathematical analysis of the signal diminishing problem and introduce a highly original TTFS-based initialization method.
2. The proposed decoder is interesting. It encourages early spikes, which provides new ideas for TTFS training.
3. The framework achieves state-of-the-art accuracy and impressively low inference time-steps on several medium benchmarks. The lack of ImageNet evaluation is acceptable given the current scaling challenges of all TTFS SNNs.
4. This article is well-written and very easy to understand.

**Weaknesses**
1. Average-pooling outputs fractional values (e.g., 0.25), violating the strict binary input constraint of AMOS neurons. The paper does not explain how subsequent layers handle these continuous inputs.
2. The proposed decoder requires exhaustive manual tuning for its decay factor ($\gamma$) and type. The paper lacks a theoretical heuristic for setting these parameters on new datasets.
3. AMOS neurons are forced to fire during training but can remain silent during inference. The paper fails to address the potential distributional shift in post-synaptic currents caused by this mismatch.
4. The paper notes that prior methods (TQ-TTFS, DTA-TTFS) fail to converge on CIFAR-100 without pre-training, but it does not theoretically explain the specific bottlenecks causing these failures.

---

> ### Author Rebuttal · Authors · 2026-03-31
>
> Thank you for your constructive feedback and for recognizing the originality of our ETTFS framework!
>
> > Q1. While average-pooling preserves the temporal sum of 1 (Eq. 24), dividing by the spatial dimension  outputs continuous, fractional values (e.g., 0.25) rather than binary {0, 1} spikes. How do subsequent layers process these non-binary inputs without violating strictly AMOS constraint?
>
>
> The non-binary, fractional values produced by average-pooling do not violate the system constraints because average-pooling and subsequent convolutional layers are both linear operations.
>
> - Mathematical Consistency: While the output $S_{\text{out}}$ at a specific time-step $t$ may be fractional (e.g., $1/n$), the temporal sum of the sequence remains strictly $1$:
>     $$\sum_{t=0}^{T-1}S_{\text{out}}[t] = \sum_{t=0}^{T-1} \frac{1}{n}\sum_{i=0}^{n-1}S_{\text{in}}[t][i] = 1$$
>     This ensures the "one-spike characteristic" is preserved in a probabilistic or weighted sense across the temporal window.
> -  Layer Fusion: Because these operations are linear, they can be theoretically fused into a single subsequent convolutional layer. In this fused implementation, the subsequent convolutional layer essentially processes a weighted version of the input spikes. Since the integration of these fractional values within the LIF neuron's membrane potential follows the same linear dynamics as binary spikes, the underlying TTFS temporal logic remains intact without requiring the inputs to be strictly binary $\\{0, 1\\}$.
>
> > Q2. The authors admit that $\gamma$ requires "dataset-specific empirical tuning," observing that an exponential decay works best for Fashion-MNIST, while linear decay works best for CIFAR10. Is there a theoretical guideline or heuristic to select these parameters for unseen datasets without exhaustive search?
>
> We acknowledge the reviewer's point regarding empirical tuning; currently, the selection of decay strategies is guided by the inherent complexity and temporal sparsity of the dataset. See Reviewer JkbG Q2 answer for more $\gamma$ selection details.
>
>
>
> > Q3. AMOS neurons are forced to fire at the last time-step during training, but permitted to remain silent during inference for early stopping. Does this structural mismatch cause a distributional shift in post-synaptic currents during evaluation, and if so, how is it mitigated?
>
> This structural mismatch does not cause a distribution shift or accuracy loss because:
> 1. the vast majority of spikes are either emitted well before the final time-step or remain entirely silent.
> 2. The model is causal; therefore, this only affects the final time-step.
>
> Consequently, forcing a spike at the final step during training is to ensure training stability and does not interfere with the learned membrane potential distributions or the inference-time early stopping mechanism.
>
> > Q4. Why do prior methods like TQ-TTFS and DTA-TTFS struggle to converge on complex datasets like CIFAR-100 without the aid of pre-trained weights?
>
> The convergence struggles of prior methods like TQ-TTFS and DTA-TTFS on complex datasets stem from their decoder designs, which fail to facilitate effective gradient backpropagation during training. In contrast, our proposed TWD addresses this by employing decayed temporal weights to prioritize early spikes, ensuring a more stable and differentiable training process. This mechanism allows for efficient gradient flow even without pre-trained weights, while maintaining the low latency and sparse binary representations inherent to TTFS coding.

---

> > ### Author Rebuttal · Reviewer_1jbn · 2026-04-03
> >
> > Thanks for the author's reply. My concerns have been addressed.

---

### Official Review · Reviewer_JkbG · 2026-03-09

**Soundness:** 3
**Presentation:** 4
**Significance:** 3
**Originality:** 4
**Overall Recommendation:** 5
**Confidence:** 4

**Summary:**

The paper proposes ETTFS, a training framework designed to address the unstable training and low accuracy issues inherent in Time-to-First-Spike (TTFS) coding. While TTFS coding is highly energy-efficient, it often suffers from training instability and low accuracy. To fix this, the authors propose ETTFS-init to stop signals from diminishing and weight normalization to keep distributions stable during training. They also add a temporal weighting decoder that reduces latency by encouraging earlier spikes, and they replace max-pooling with average-pooling to maintain the strict one-spike constraint. The framework achieves state-of-the-art accuracy on multiple benchmarks.

**Compliance With Llm Reviewing Policy:**

Affirmed.

**Final Justification:**

My concerns are solved, so I am inclined to keep my original score.

**Key Questions For Authors:**

1. While Figure 2 illustrates the physical latency differences between layer-by-layer and step-by-step propagation, could you explicitly clarify the algorithmic or implementation differences in the forward and backward passes between these two paradigms? How should early stopping be implemented?
2. The experiments are limited to small values of the hyperparameter γ. It would be informative to explore larger values (e.g., > 3) and report their effect on accuracy to better understand the sensitivity of the model to this parameter.
3. E3 is calculated using a single, idealized convolutional layer in Spikesim (262 M-neurons, 256 channels, 3 $\times$ 3 kernel, 32 $\times$ 32 feature map). Does this localized benchmark generalize to the end-to-end memory access energy of the entire network?

**Limitations:**

yes

**Strengths And Weaknesses:**

### Strengths

1. **Theoretical Grounding:** The authors propose the ETTFS initialization and weight normalization based on rigorous mathematical analysis to effectively mitigate signal diminishing and gradient vanishing.
2. **Strong Benchmark Performance:** Experimental and comparative results across multiple datasets demonstrate the effectiveness and performance–inference latency advantage of the proposed framework.
3. **Well‐Presented:** The manuscript is well‐structured and clearly written. The figures are informative and effectively support the presentation of the methods and results.

### Weaknesses

1. **Unclear Algorithmic Details:** The paper lacks explicit algorithmic explanations for the step-by-step and layer-by-layer propagation of differentiating forward and backward passes. Additionally, the exact implementation mechanics of early stopping are not described.
2. **Limited Hyperparameter Analysis:** The proposed decoder is only evaluated with small values of the decay factor $\gamma$. Exploration of larger values (e.g., $\gamma > 3$) is needed to properly establish the model's sensitivity and robustness.
3. **Narrow Scope of the E3 Metric:** Memory access energy (E3) is calculated using only a single, idealized convolutional layer scenario. It remains unclear whether this localized benchmark accurately reflects the end-to-end memory energy consumption of the entire network.

---

> ### Author Rebuttal · Authors · 2026-03-31
>
> We thank the reviewer for the insightful questions and for the high assessment of our paper's soundness and presentation.
>
> > W1 & Q1. While Figure 2 illustrates the physical latency differences between layer-by-layer and step-by-step propagation, could you explicitly clarify the algorithmic or implementation differences in the forward and backward passes between these two paradigms? How should early stopping be implemented?
>
> Our ETTFS applies a causal model; its layer-by-layer and step-by-step propagations are mathematically equivalent, allowing causal models to leverage GPU efficiency during training and low-latency neuromorphic hardware during inference. Our work addresses the performance gap where causal models traditionally underperform their non-causal counterparts, maintaining high accuracy alongside temporal flexibility. Refer to Reviewer 3QsV Q2 for more details.
>
> - Propagation Paradigms and Algorithmic Differences
> In our case, layer-by-layer and step-by-step propagation are mathematically equivalent but hardware-distinct. Layer-by-layer treats time as a tensor dimension, optimizing GPU throughput by computing $\mathbf{O}\_{1:T}^{(l)} = f(W^{(l)}, \mathbf{O}\_{1:T}^{(l-1)}, \mathbf{V}\_{0}^{(l)})$ per layer. Conversely, step-by-step minimizes memory by iterating through time steps $t \in \{1, \dots, T\}$ across all layers: $O\_t^{(l)} = f(W^{(l)}, O\_t^{(l-1)}, V_{t-1}^{(l)})$, enabling lower physical latency. For training, both use layer-by-layer-based BPTT to calculate gradients via surrogate functions: $\frac{\partial \mathcal{L}}{\partial W^{(l)}} = \sum_{t=1}^{T} \frac{\partial \mathcal{L}}{\partial O\_t^{(l)}} \cdot \sigma'(V\_t^{(l)}) \cdot \frac{\partial V\_t^{(l)}}{\partial W^{(l)}}$.
>
> - Implementation of Early Stopping
> In ETTFS, early stopping reduces energy by terminating simulation at $T\_{\text{stop}} = \min \\{t \mid \sum_{i} O\_{t,i}^{\text{out}} > 0 \\}$. Once the decoder layer detects the first output spike at $t$, the predicted class is identified as $Y\_\text{predict} = \text{argmin}\_{i}\\{t|O[t][i] = 1\\}.$. In step-by-step mode, hardware ceases operations immediately at $T\_{\text{stop}}$, bypassing remaining steps. ETTFS specifically optimizes this by concentrating spike information into earlier intervals, significantly lowering average latency compared to standard TTFS.
>
>
>
> > Q2. The experiments are limited to small values of the hyperparameter γ. It would be informative to explore larger values (e.g., > 3) and report their effect on accuracy to better understand the sensitivity of the model to this parameter
>
> We investigated the impact of increasing the decay factor $\gamma$ beyond 3 and observed that the resulting performance does not exhibit a linear trend, as illustrated in the following figure.
>
> Table R1. Fashion-MNIST with Exponential Decay Function
>
> |**γ**|**3**|**4**|**5**|**6**|**7**|
> |---|---|---|---|---|---|
> |**Accuracy (%)**|92.90|92.40|91.45|87.72|78.73|
>
> Experimental results indicate that the optimal range for the $\gamma$ parameter is between 2 and 3, beyond which accuracy begins to degrade.
>
>
> > Q3. E3 is calculated using a single, idealized convolutional layer in Spikesim (262 M-neurons, 256 channels, 3 $\times$ 3 kernel, 32 $\times$ 32 feature map). Does this localized benchmark generalize to the end-to-end memory access energy of the entire network?
>
> We appreciate the reviewer's insightful observation regarding the scope of the $E3$ metric. Although $E3$ is calculated using a standardized convolutional layer configuration, this localized benchmark effectively generalizes to the end-to-end memory energy consumption of the entire network, owing to the additive nature of energy and the event-driven hardware mechanics inherent to SNNs.

---

> > ### Author Rebuttal · Reviewer_JkbG · 2026-04-01
> >
> > My concerns have been adequately addressed. I will keep my score.

---

### Official Review · Reviewer_CobL · 2026-03-12

**Soundness:** 3
**Presentation:** 3
**Significance:** 2
**Originality:** 3
**Overall Recommendation:** 4
**Confidence:** 2

**Summary:**

This paper proposes a new training framework for training spiking neural networks with time-to-first-spike coding. Specifically, their method considers parameter initialization and normalization. Empirically, the authors demonstrate the effectiveness of their method on several ML datasets.

**Compliance With Llm Reviewing Policy:**

Affirmed.

**Key Questions For Authors:**

1. Are there multiple runs for the experiments?
2. In Table 4, can the authors report the results when only using ETTFS-init?

**Limitations:**

The limitations are discussed in section 5.

**Strengths And Weaknesses:**

### Strengths

1. The paper is well-written and easy to follow. The derivations and figures are clear enough for me to understand.
2. The proposed method is inspired by both theory and empirical observations, thus sound.
3. To my best knowledge, the proposed initialization is novel.


### Weaknesses

1. I'm unsure about how significant the empirical results are statistically as no standard deviation is being reported.


Overall, while the paper seems reasonable to me, I am not and expert in the field to understand the significance of the proposed method.

---

> ### Author Rebuttal · Authors · 2026-03-31
>
> We thank Reviewer CobL for recognizing the novelty of our proposed method and the clarity of our manuscript.
>
> > W1 & Q1. Are there multiple runs for the experiments?
>
> Each result represents the average of multiple independent runs, with minimal variance observed between trials. We will include standard deviations in the revised manuscript.
>
> > Q2. In Table 4, can the authors report the results when only using ETTFS-init?
>
> We sincerely thank the reviewer for this suggestion. We have updated Table 4 to include the classification accuracy when utilizing only ETTFS-init, allowing for a clearer ablation study. The revised table is provided below:
>
> R1. Evaluating the components of ETTFS on the Fashion-MNIST dataset.
>
> | ETTFS-init | Avg-pooling | Norm | Norm with affine | TWD | Accuracy   |
> | ---------- | ----------- | ---- | ---------------- | --- | ---------- |
> |            |             |      |                  |     | 89.61%     |
> |            |             |      |                  | ✓   | 89.72%     |
> | ✓          |             |      |                  |     | 90.27%     |
> | ✓          |             |      |                  | ✓   | 90.47%     |
> |            | ✓           |      |                  | ✓   | 90.41%     |
> | ✓          | ✓           |      |                  | ✓   | 90.88%     |
> | ✓          | ✓           | ✓    |                  | ✓   | 91.61%     |
> | ✓          | ✓           |      | ✓                | ✓   | **92.90%** |
>
> If you have any further questions, please do not hesitate to ask.

---

> > ### Author Rebuttal · Reviewer_CobL · 2026-04-03
> >
> > The authors addressed my questions well.

---

### Official Review · Reviewer_3QsV · 2026-03-13

**Soundness:** 4
**Presentation:** 4
**Significance:** 3
**Originality:** 3
**Overall Recommendation:** 5
**Confidence:** 4

**Summary:**

In this papers, the authors propose a slew of methods for training SNNs with time-to-first-spike coding (TTFS), including a new initialisation method, weight normalisation, a new decoder and the observation that average pooling is more suited for TTFS. The authors train SNNs with TTFS applying these methods on various tasks, estimate energy consumption and perform ablation studies.

**Compliance With Llm Reviewing Policy:**

Affirmed.

**Final Justification:**

The authors addressed most of my concerns in their response. I'll maintain my high score.

**Key Questions For Authors:**

1. I assume the networks are trained with surrogate gradient. Could you clarify if this is the case (in the paper as well)?
2. In section 2.1, you say that the layer-by-layer and causal step-by-step implementations are mathematically equivalent, the latter achieves lower accuracy. I would like to understand why this is the case. Could you elaborate on that?
3. In section 3.4, you say "The temporal nature of TTFS coding suggests decoding through the earliest spike timing." Isn't that by definition, or is there other ways of using TTFS? I found this part a bit confusing, and wasn't clear what argument was being made there.
4. How does the temporal weighting decoder compare to the one in (Wunderlich & Pehle 2021) where they do a softmax over the spike times?

### References:

Wunderlich, Timo C., and Christian Pehle. 2021. ‘Event-Based Backpropagation Can Compute Exact Gradients for Spiking Neural Networks’. Scientific Reports 11 (1): 12829. https://doi.org/10.1038/s41598-021-91786-z.

**Limitations:**

The authors have briefly discussed the limitations of their method in Sec 5.

**Strengths And Weaknesses:**

## Strengths

- Training of TTFS SNNs is very timely since TTFS has the potential to improve energy utilisation significantly. The authors work clearly solves many of the challenges associated with TTFS.
- The reasoning for all the introduced methods is sound, and principled.
- The empirical evaluation is done well. The authors test their methods on feed-forward models on tasks ranging from MNIST to CIFAR100, and DVS Gesture. The models tested are fairly large models.
- The fact that ETTFS achieves very low time-steps to solution during inference is impressive.
- The methods are reasonably well explained, and structured well.
- Many of the proposed changes are novel to my knowledge.

## Weaknesses

- The main weakness of the paper (as the authors themselves point out) is that they do no test on larger scale tasks such as Imagenet.
- It would also be interesting to see how well these methods fare when applied to sequence models (spiking RNNs/transformers).
- It is not clear if the reported results are over multiple runs or single runs. If multiple runs, the standard deviation should also be reported.
- non-TTFS baselines are not provided (to see how big the gap still is between TTFS and non-TTFS methods)

### Minor:
- Section 4.1.1 "relie on" -> "reliant on"?
- Table 3: TWD -> ETTFS?

---

> ### Author Rebuttal · Authors · 2026-03-31
>
> We thank the reviewer for recognizing our work's novelty and performance. Your feedback is invaluable for improving the paper.
>
> > W1. No test on Imagenet.
>
> We must honestly acknowledge that ETTFS currently cannot be trained on large-scale tasks like ImageNet. However, we believe the proposed ETTFS provides a foundation for future research addressing the fundamental 'zero-to-one' challenge of achieving convergence when directly training TTFS. Future studies can follow improve this framework to achieve better performance on ImageNet.
>
> > W2. Applied to sequence models like (spiking RNNs/transformers).
>
> The proposed ETTFS framework is derived from the LIF sequential model; therefore, it is fundamentally a sequence-based architecture.
>
> Regarding the Transformer, our current work focuses on convolutions because integrating TTFS with attention is challenging. However, we may explore adapting ETTFS concepts for Transformers in future research.
>
>  > W3. Multiple runs?
>
> Each result represents the average of multiple independent runs, with minimal variance observed between trials. We will include standard deviations in the revised manuscript.
>
> > W4. Non-TTFS baselines.
>
> To facilitate a clearer comparison of the performance gap with rate coding, we have followed the reviewer's suggestion and included the classification accuracy for rate coding within the main text:
>
> Table R1. Rate Coding Comparison
>
> | Method | MNIST | Fashion-MNIST | CIFAR10 | CIFAR100 | DVSGesture |
> | --- | --- | --- | --- | --- | --- |
> | Rate | FC400 / 99.54% | SCNN / 93.21% | SCNN / 94.29% | SCNN / 73.21% | SCNN / 93.47% |
> | ETTFS (Ours) | SCNN / 99.48% | SCNN / 92.90% | SCNN / 93.89% | SCNN / 72.7% | SCNN / 95.83% |
>
> With identical architectures, ETTFS shows only marginal gaps compared to rate coding on CIFAR-10/100, while significantly improving energy efficiency and even surpassing rate-coding performance on DVS datasets.
>
> > Minor error
>
> We sincerely appreciate the reviewer for identifying these two typo errors; we have corrected them in the revised manuscript.
>
> > Q1. I assume the networks are trained with surrogate gradient.
>
> As noted, ETTFS adapts the LIF sequential model and employs surrogate functions for training; we have added this detail to the Experiments section per the reviewer’s suggestion.
>
> > Q2. Layer-by-layer and causal step-by-step confusion.
>
> We apologize for the lack of clarity in Section 2.1. We have provided a more detailed explanation below and have incorporated these clarifications into the revised manuscript:
>
> TTFS SNNs are implemented through two propagation paradigms:
> - Non-Causal Models: These models are restricted to layer-by-layer execution. A prominent example is the ANN2SNN method, which lacks the temporal dependencies required for incremental, step-by-step propagation.
> - Causal Models: These frameworks, such as TQ-TTFS and the proposed method, support both layer-by-layer and step-by-step propagation. In causal models, these two modes are mathematically equivalent, ensuring consistent output regardless of the execution strategy. Furthermore, layer-by-layer propagation offers training efficiency on GPUs, whereas step-by-step propagation minimizes latency on neuromorphic hardware.
>
> Despite the flexibility of causal models, existing methods suffer from lower performance compared to their non-causal counterparts. This work attempts to address the accuracy limitation while maintaining the low latency of the causal model.
>
>
> > Q3. "The temporal nature of TTFS coding suggests decoding through the earliest spike timing." confusion.
>
> TTFS has only one way to express. We emphasize the earliest spike timing in TTFS coding to demonstrate that achieving this property requires implementing a non-linear, non-differentiable operation at the output decoder layer:
> $$Y_\text{predict} = \text{argmin}_{i}\{t~|~O[t][i] = 1\}.$$
> Since this formulation is inherently non-differentiable and thus unsuitable for direct gradient-based training, we propose a novel decoder to address this limitation.
>
> > Q4. Softmax over the spike times?
>
> We appreciate this insightful suggestion. However, our preliminary experiments indicated that applying a standard Softmax over the temporal dimension is infeasible. The rationale is that our temporal weighting decoder is predicated on the assumption that an earlier spike must be assigned a strictly higher weight than a subsequent one. Without additional weighting parameters, a standard softmax fails to satisfy this condition, leading to a failure to converge.
>
> To address this, we adapt a softmax operation to the membrane potentials during training. By selecting the maximum membrane potential to during inference, we achieved competitive results, as summarized in the table below.
>
> Table R2 : Fashion-MNIST Performance Comparison
>
> |Method|Accuracy (%)|Observations|
> |---|---|---|
> |Softmax (Temporal)|—|Failed to converge|
> |Softmax (Membrane)|92.04%|Comparable performance|
> |ETTFS (Ours)|92.90%|Superior performance|

---

> > ### Author Rebuttal · Reviewer_3QsV · 2026-04-01
> >
> > All my concerns have been fully addressed. I will maintain my high score.

---

### Decision · Program_Chairs · 2026-04-30

**Decision:**

Accept (spotlight)

**Comment:**

This paper presents a timely and important contribution to training Time-To-First-Spike (TTFS) spiking neural networks, a paradigm with potential for improved energy efficiency. The work addresses several key challenges in TTFS training with solutions that are both principled and well-motivated.

Reviewers emphasize that ETTFS achieves very low time-steps to solution during inference. Empirical evaluations are somewhat modest, based on modern expectations, but satisfied the reviewers. The paper is well-structured and clearly written. Overall, the methods appear novel and the work makes a meaningful contribution to energy-efficient neuromorphic learning, which will be interesting for the ICML community. I recommend to accept.